# Temperate functional niche availability not resident-invader competition shapes tropicalisation in reef fishes

Mark G. R. Miller[1,2] ✉, James D. Reimer [3,4], Brigitte Sommer [5,6],
Katie M. Cook [1,7], John M. Pandolfi [8], Masami Obuchi[4,9] & Maria Beger [1,10] ✉

Temperate reefs are at the forefront of warming-induced community alterations resulting from poleward range shifts. This tropicalisation is exemplified and amplified by tropical species' invasions of temperate herbivory functions. However, whether other temperate ecosystem functions are similarly invaded by tropical species, and by what drivers, remains unclear. We examine tropicalisation footprints in nine reef fish functional groups using trait-based analyses and biomass of 550 fish species across tropical to temperate gradients in Japan and Australia. We discover that functional niches in transitional communities are asynchronously invaded by tropical species, but with congruent invasion schedules for functional groups across the two hemispheres. These differences in functional group tropicalisation point to habitat availability as a key determinant of multi-species range shifts, as in the majority of functional groups tropical and temperate species share functional niche space in suitable habitat. Competition among species from different thermal guilds played little part in limiting tropicalisation, rather available functional space occupied by temperate species indicates that tropical species can invade. Characterising these drivers of reef tropicalisation is pivotal to understanding, predicting, and managing marine community transformation.

Unravelling the unknown mechanisms of species redistribution and resulting community transformations is vital for maintaining coral, subtropical, and temperate reef biodiversity and ecosystem services, valued at USD 16.7 trillion annually[1]. Warming-induced community transformation[2] is the result of differential species responses to changing and novel climatic conditions[3]. 'Tropicalisation' denotes a process where poleward shifts of tropical species, such as corals and tropical herbivorous fishes, alter the character of a marine ecosystem from algae-dominated to coral-dominated habitat structures and associated fauna[4–6]. Tropicalisation impacts are currently understood for few species and functions (e.g., fish herbivory[7–10], but see[11] for consideration of other trophic groups) and tropicalisation drivers are typically explored at regional[9,12] or single species level[13]. However, to comprehend and predict the occurrence and consequences of tropicalisation, a community perspective across space and time is required[14].

[1]School of Biology, Faculty of Biological Sciences, University of Leeds, Leeds LS2 9JT, UK. [2]School of Biological Sciences, Monash University, Clayton, VIC 3800, Australia. [3]Graduate School of Engineering and Science, University of the Ryukyus, 1 Senbaru, Nishihara, Okinawa 903-0213, Japan. [4]Tropical Biosphere Research Center, University of the Ryukyus, 1 Senbaru, Nishihara, Okinawa 903-0213, Japan. [5]School of Life and Environmental Sciences, The University of Sydney, Sydney, NSW 2006, Australia. [6]School of Life Sciences, University of Technology Sydney, Sydney, NSW 2007, Australia. [7]National Institute of Water and Atmosphere Research, Hamilton, New Zealand. [8]Australian Research Council Centre of Excellence for Coral Reef Studies, School of Biological Sciences, The University of Queensland, Brisbane, QLD, Australia. [9]Endo Shell Museum, 1175 Manatsuru, Ashigarashimo-gun, Manazuru-machi, Kanagawa 259-0201, Japan. [10]Centre for Biodiversity Conservation Science, School of Biological Sciences, The University of Queensland, Brisbane, QLD, Australia. ✉e-mail: mark.gr.miller@gmail.com; m.beger@leeds.ac.uk

Species within a community with similar trait combinations can be assigned to functional groups, which can represent ecosystem functions[15,16]. Trait-based approaches can therefore help understand the functional processes associated with community re-organisation from tropicalisation[10,17], and which species are likely to advance poleward[12,18]. Traits can also define a species' functional niche, and akin to Hutchinson's definition of the ecological niche as 'the volume in multi-dimensional environmental space where stable populations can be maintained'[19], we define the functional niche as the multi-dimensional trait space that contains species with similar traits[20]. Specifically, the trait space occupied by all species within a functional group represents the breadth of functional roles the functional group has the ability to perform (fundamental functional niche). In any given community, the breadth of these functional roles (i.e., the realised functional niche) is constrained by local abiotic and biotic pressures that limit which species locally occur[20,21].

Within each functional group, changes in the functional niches occupied by tropical and temperate species delineate potential mechanisms driving tropicalisation. A contraction of functional niche area can serve as a measure of environmental filtering[18], and is expected in tropical fishes from low to high latitudes as thermal tolerance ultimately determines their survival[22,23]. Tropical species with functional niches tied to specific habitat, such as reef-forming corals, may exhibit limited poleward advance[11], whereas tropical 'generalists' that have functional niches with broad habitat and/or dietary requirements, have greater arrival and establishment levels in higher latitudes[12,24,25]. However, the effects of biotic interactions within functional niches on tropicalisation are less well known (but see ref. 26), and in particular whether competition with resident temperate species limits poleward advances in tropical species[27].

Darwin's naturalisation hypothesis attributes an advantage to invasive species that minimise competition with residents by being different[28]. For example, using species overlaps in trait space to infer competition, Azzurro et al.[29] find tropical fish invading the Mediterranean that display different morphological niches to those of resident species establish abundant populations, whereas those that shared morphological trait spaces do not. In contrast, using the same method, Smith et al.[30] find increasing occurrence of tropical fishes does not relate to the uniqueness of their morphological niche when invading an Australian temperate kelp community.

Multiple abiotic and biotic drivers of niche availability likely affect functional groups differently, and we might expect differential rates of tropicalisation among them[11,27]. The degree to which some temperate functional niches are more easily invaded by tropical species than others, and the interactions among functional groups during tropicalisation (e.g., tropical herbivores facilitating corallivore colonisation through kelp-to-coral phase shifts), determine whether tropical species, and their respective functions, arrive at higher latitudes together or mismatched in time[14]. Along those lines, Darwin's pre-adaptation hypothesis contrasts with his naturalisation hypothesis, and attributes an advantage to invasive species that share traits with residents, pre-adapting them to their new environment[31]. This hypothesis would support the potential lack of competition between resident and incoming tropicalising species. Here, we evaluate this unresolved process to address its important implications for the capacity of recipient communities to resist tropicalisation and maintain original ecosystem functions[17].

Here, we compare tropicalisation and functional niches represented in trait space across major functional groups in reef fish communities from biogeographical transition zones of Australia and Japan. Both regions are well-known, but contrasting, tropicalisation hotspots; their different warming patterns and projected future species gains and losses[32] therefore allow broad generalisation of our findings. First, we ask whether functional groups are synchronously or asynchronously invaded by tropical species in each region. Second, we assess whether competition (as inferred by functional niche overlap) between tropical and temperate thermal guilds is prevalent across functional groups. Last, we compare poleward leading edges of tropical species biomass in each functional group, and combine regions to identify drivers of tropicalisation differences among groups. Characterising these drivers of reef tropicalisation will be pivotal to understanding and predicting marine community transformation.

## Results

### Characterising major trait-based functional groups of tropical-temperate reef fish communities

We define six transitional zones, or fish community types (Fig. 1), with hierarchical clustering of fish biomass from tropical to temperate environmental gradients. Transitional zones represent distinct community turnover between island groups or available habitat structure across the environmental gradient, driven primarily by latitude, sea surface temperature and coastal geography (PERMANOVA, Supplementary Fig S1).

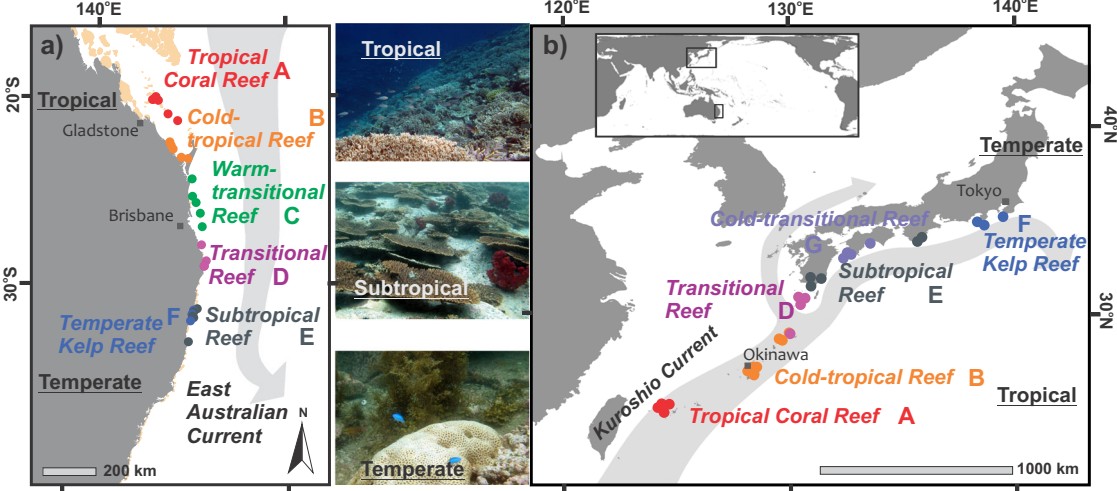

**Fig. 1 | Map of tropicalising reef zones.** Survey sites (dots) grouped into transitional community zones (coloured text, for analysis of environmental space, see Supplementary Fig. S1) using fish biomass in: (**a**) Australia and (**b**) Japan. Peach shade in Australia shows the southern extent of Great Barrier Reef. Photographs by M Beger and B Sommer.

We discover that the reef fish meta-community from tropical-temperate transitional zones of Australia and Japan is organised into nine major functional groups (FG). The nine functional groups represent 85% of species in the meta-community (additional ten rarer functional groups complete the meta-community; Supplementary Fig. S2) that we identify using hierarchical clustering of five traits (diet, habitat association, body size, aggregation, and depth range, Supplementary Table S1) linked to species effects on ecosystem functioning[33,34]. Functional group delineation is driven mostly by the trait diet (51.3%−variable importance from multinomial random forest[35]), closely followed by habitat association (46.3%), with body size (1.4%), aggregation (0.8%) and depth range (0.2%) contributing much less (Fig. 2 and Supplementary Fig. S2). Functional groups are therefore named using diet and habitat association traits (green and red in Fig. 2c), with the remaining traits describing functional niche details. A final trait, thermal guild, is used to assign species within functional groups as either tropical or temperate; a comparison of the realised upper thermal limits of tropical species[36] confirmed functional groups are independent of thermal preference (Supplementary Fig. S5). All functional groups but one (Corallivores FG16) include tropical and temperate species, indicative of temperate functional niches that tropical species could invade. The trait combinations of species in the meta-community are organised into functional groups of varying species richness (Fig. 2). We consider ecosystem functions performed by many species to represent 'generalist' functional groups (e.g., Benthic Predators FG15), whereas 'specialist' functional groups perform ecosystem functions that accommodate fewer species (e.g., Demersal Predators FG12)[37]. Total fish biomass is a widely used indicator of reef status and functioning[38–40], and the biomass of functional trophic groups can describe the pressure of their ecosystem function[41–43]. We find that the biomass of tropical species declines with increasing latitude within all but one of the nine functional groups in both regions (Fig. 2). The biomass of temperate species within each functional group generally shows the opposite trend. Functional pressure (total biomass) is generally maintained across latitude in functional groups with comparable tropical and temperate biomass, but several functional groups are dominated by tropical biomass, and their functional pressure declines with increasing latitude (e.g., Upper-benthic Herbivores FG6).

## Asynchronous functional group tropicalisation

We find that functional group tropicalisation footprints differ significantly in every transitional zone but show consistencies between Australia and Japan in their change over latitude (Fig. 3). Tropicalisation is characterised as the proportional increase of tropical species in communities over time[44]. We substitute space for time by calculating the 'tropicalisation footprints' (proportion of biomass of tropical species relative to that of the Tropical Coral Reef zone) of communities over a latitudinal gradient. By assigning tropical species to functional groups, we compare tropicalisation of ecosystem functions by assessing how well functional group tropicalisation footprints track community-level tropicalisation footprints over latitude, and at what latitude each functional group shows a 20-fold decline in biomass (defined as their 'tropicalisation leading-edge'). Community-level biomass of all tropical species declined significantly with increasing latitude in both regions (ANOVA, Australia, $F_{(5,19)} = 8.46$, $P = < 0.001$; Japan, $F_{(5,23)} = 11.18$, $P = < 0.001$; Fig. 3), but showed greater poleward advance in Japan than Australia (Supplementary results).

Corallivores FG16, Benthic Planktivores FG4 and Upper-benthic Herbivores FG6 were the least tropicalised functional groups in both regions. Their tropicalisation footprints scored significantly below the community-level expectation in multiple zones (Fig. 3), and their tropicalisation leading edges did not extend as far poleward as the community expectation (terminating in zones B, D & E; Table 1). Demersal Predators FG12 and Upper-benthic Omnivores FG1 were less

tropicalised in Japan but tracked community tropicalisation in Australia. Upper-benthic Planktivores FG4, Benthic Herbivore/Omnivores FG8, Benthic Predators FG15 and Upper-benthic Predators FG10 constituted the most tropicalised functional groups in both regions. Their tropicalisation footprints tracked, or in some zones significantly exceeded, the community-level expectation and their tropicalisation leading edges extended into high-latitude Temperate Kelp Reef (zone F; Table 1).

The known tropicalisation of herbivore functions in transitional communities[7,8] is supported by patterns in Benthic Herbivores/Omnivores FG8, which show higher tropicalisation footprints than the community expectation in mid-latitude zones (zones B, C and D; Fig. 3) and tropicalisation leading edges at high latitudes (Table 1). Our results indicate that the ecosystem function performed by tropical Benthic Herbivore/Omnivores exerted greater pressure (greater biomass) in Japan's Cold-tropical Reef zone, and Australia's Warm-transitional Reef zone (~27° N/S; Fig. 3) than in Tropical Coral Reef zones. However, contrary to expectations, Upper-benthic Herbivores FG6 showed lower tropicalisation footprints than the community expectation, particularly in Australia. This finding illustrates important ecological differences between Benthic and Upper-benthic Herbivores (FG8 vs FG6, Supplementary Fig. S2C) likely related to schooling, behaviour, and habitat requirements. Our results suggest that, with our two herbivore functional groups spanning low and high tropicalisation rates, existing herbivore studies may be broadly representative of fish community tropicalisation. However, our results support a careful choice of herbivore species with which to infer community tropicalisation; for example, Upper-benthic Herbivores in Australia were consistently less tropicalised than the community. Given that herbivory facilitates tropicalisation in eastern Australia[7], these processes likely illustrate the importance of invertebrate herbivory in concert with fish grazing[45]. Failure to consider herbivore species included in analyses may lead to inadvertent generalisation or bias and miss important aspects of community tropicalisation.

Our results complement well-studied herbivore tropicalisation by filling knowledge gaps on how the other major functional groups in transitional fish communities are tropicalised. For example, tropical predators generally keep pace with community tropicalisation (Fig. 3), and could play an important role in mediating runaway tropical herbivory pressure on temperate reefs, particularly when resident predators are reduced[45].

## Functional niche competition does not limit tropicalisation

We find limited support that thermal guild competition affects tropicalisation within functional groups (Fig. 4 and Table 1). Although tropical species show increasing functional niche overlap as they encounter increasing temperate biomass (Kendall, all functional groups $P < 0.05$, except Upper-benthic Herbivores FG6 in Australia and Upper-benthic Omnivores FG1 in Japan; Figs. 2 and 4), this overlap rarely correlates with tropicalisation footprints (Fig. 4 and Table 1). Only Upper-benthic Predators FG10 in Japan and Australia, and Benthic Herbivores/Omnivores FG8 in Japan exhibit a significant correlation between decreasing tropical biomass and increasing functional niche overlap, supporting competition[29]. By contrast, Benthic Planktivores FG4 and Benthic Predators FG15 in Japan show significant correlation between increasing tropical biomass and increasing functional niche overlap with temperate residents, suggesting resource abundance offsetting potential competition[30].

Instead, we discover that environmental filtering acts upon almost all functional groups. With the exception of Benthic Predators FG15 in Japan and Demersal Predators FG12 in Australia, tropicalisation footprints are significantly positively correlated with functional niche areas within functional groups (Fig. 4 and Table 1). As tropical biomass declines poleward, tropical species are filtered, contracting tropical

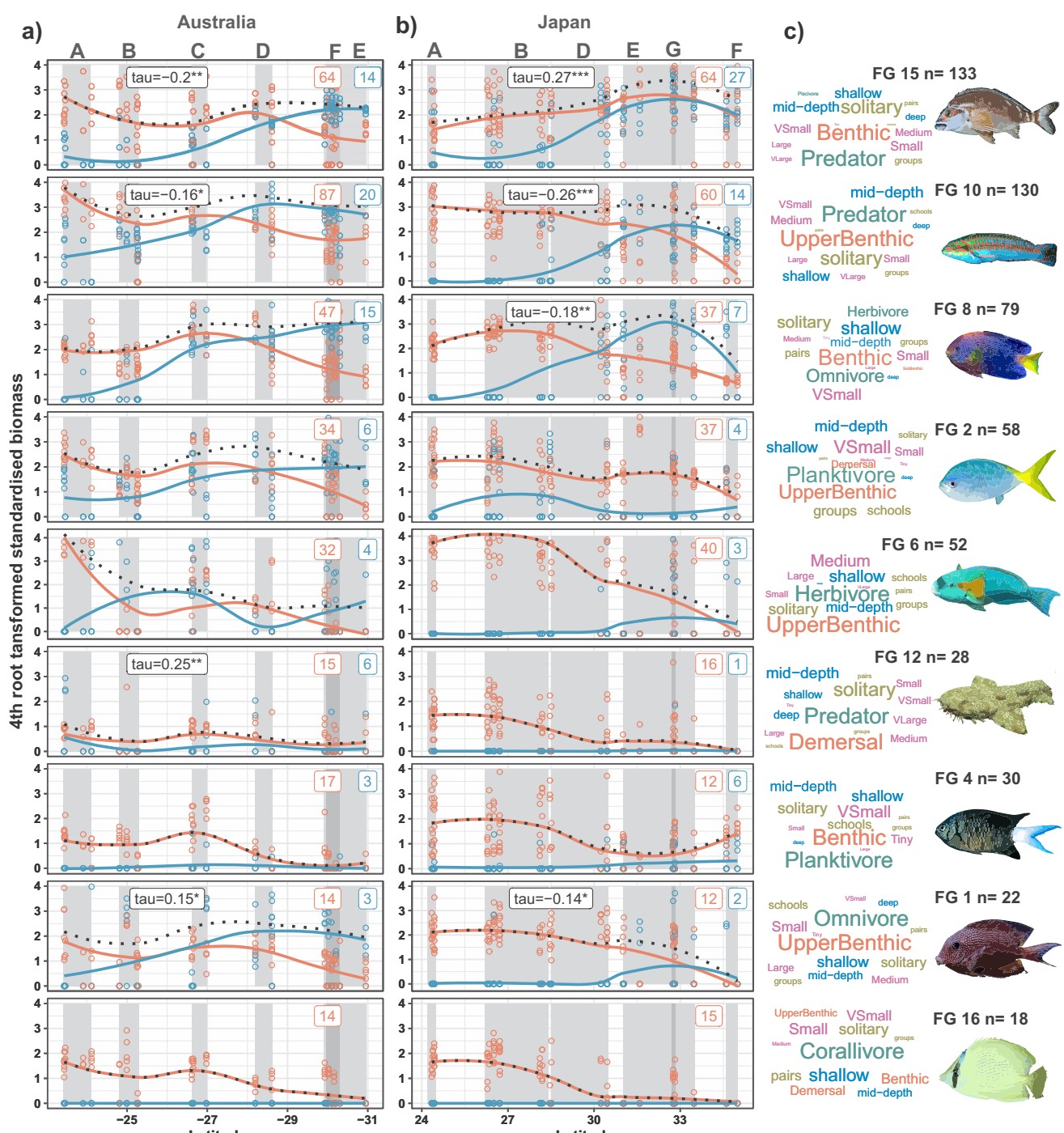

**Fig. 2 | Changes in functional group biomass over latitude.** Biomass trends of tropical (red) and temperate species (blue) in (**a**) Australia and (**b**) Japan, from (**c**) different functional groups. Loess smoothers visualise trends over latitude for each thermal guild and total biomass (dotted). Grey shade represents latitudinal spans of six zones (A–G) in each region, see Fig. 1. Inset boxes within each latitude plot denote number of tropical and temperate species from each functional group within each region. Functional groups with significant correlation between tropical and temperate biomass have Kendall's tau and associated significance shown (<0.001 = ***, <0.01 = **, <0.05 = *). Biomass outlier data points >4 omitted from plot to aid visualisation but loess curves and correlation still fitted on full data. Word clouds denote the traits characterising each functional group, where larger word size shows higher importance of a trait in defining the group, colours denote traits: diet = green; habitat association = red; body size = pink; aggregation = yellow; and depth range = blue. Fish icons highlight a typical member of each functional group. Transitional community zones are coded: A = Tropical Coral Reef, B = Cold-tropical Reef, C = Warm-transitional Reef, D = Transitional Reef, E = Subtropical Reef, F = Temperate Kelp Reef, G = Cold-transitional Reef. Fish icons depict species characteristics for the functional groups, created in CorelDraw 16 from original photographs by the authors. Source data are provided as a Source Data file.

functional niche area (best demonstrated by Corallivores FG16; Fig. 4). Benthic Predators FG15 in Japan provide the only example of increasing tropical biomass at higher latitude, and demonstrate no contraction of functional niche (Fig. 4).

Our results support limited competitive interaction between tropical and temperate species within functional niches observed previously in Australia[30,46], and confirm this trend is prevalent across functional groups in tropicalising communities. Our metric

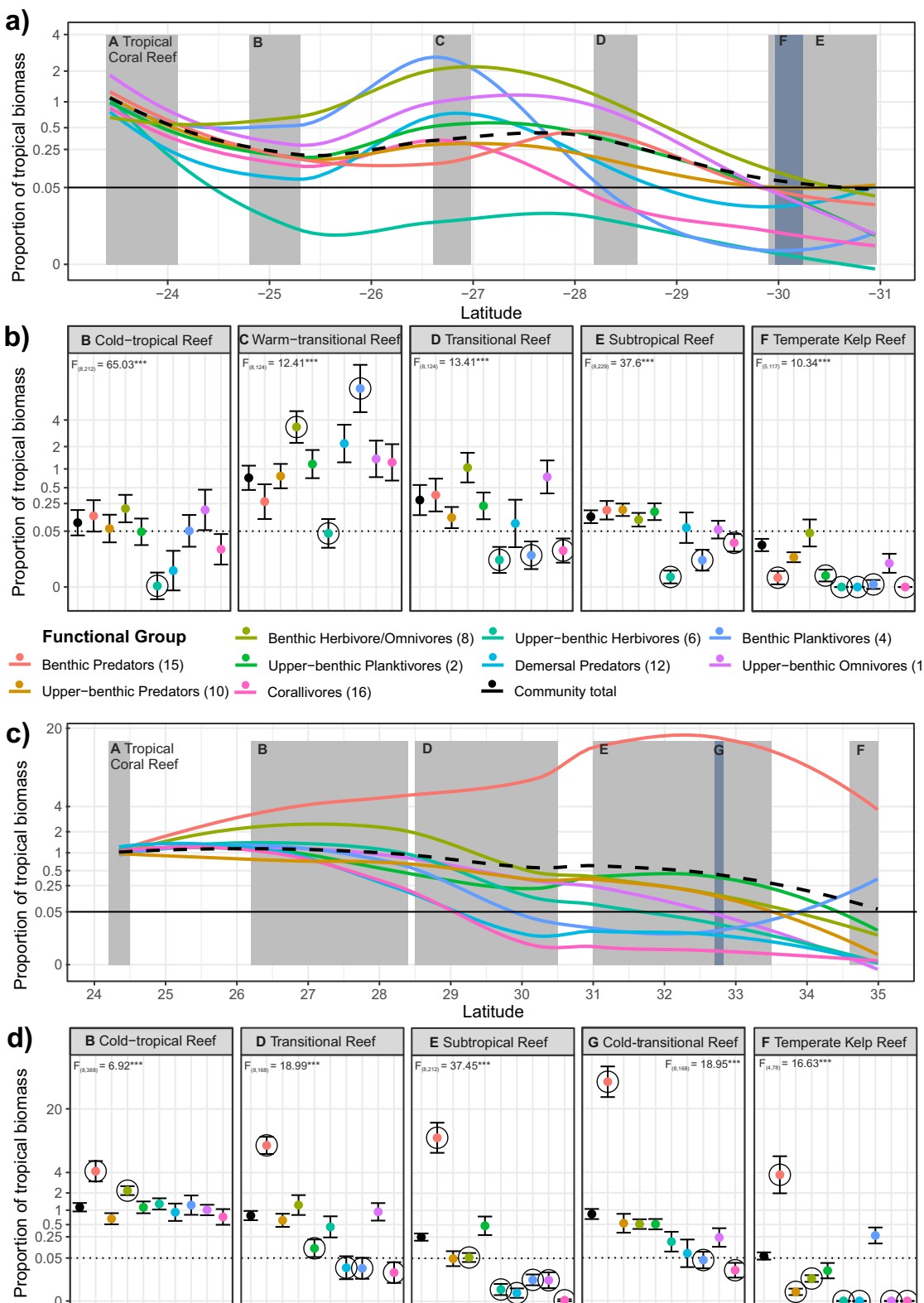

**Fig. 3 | Tropicalisation footprints in fish communities.** Tropicalisation footprints (proportion of tropical biomass relative to that of the Tropical Coral Reef zone) in Japan (**a**, **b**) and Australia (**c**, **d**) for nine functional groups (coloured) and the community total (black; dashed). **a**, **c** show change in tropicalisation footprints across latitude estimated by Loess smoothers fitted to transect level data. **b**, **d** show tropicalisation footprints (mean ± s.e.) estimated by linear mixed models within each zone, significant departure (α < 0.05) of functional group tropicalisation footprints from the community expectation are shown as circled points. Proportions are fourth-root transformed, but on *y* axes the values are back-transformed to aid interpretation; dotted lines show threshold of a 20-fold reduction in biomass. Grey shade shows the latitudinal span of named zones. Source data are provided as a Source Data file.

**Table 1 | Summary of functional group tropicalisation and functional niches in both regions**

| Functional group | Australia | | | | | Japan | | | | |
|---|---|---|---|---|---|---|---|---|---|---|
| | Tropicalisation leading-edge latitude and zone (°S) | Max tropical biomass, latitude (°S) and zone | Max temperate biomass, latitude (°S) and zone | Mean tropical-temperate functional niche overlap (%) | Mean tropical functional niche area (%) | Tropicalisation leading-edge latitude and zone (°N) | Max tropical biomass, latitude (°N) and zone | Max temperate biomass, latitude (°N) and zone | Mean tropical-temperate functional niche overlap (%) | Mean tropical functional niche area (%) |
| Benthic Predators (FG15) | 30.3 (F)' | 42.6 ± 15.4 (24.1, A) | 29.5 ± 5.7 (30.3, F) | 29 ± 22 | 62 ± 19 τ = 0.33* | Does not decline | 118 ± 31.5 (32.8, E) | 73.5 ± 16.3 (32.8, E) | 43 ± 26 τ = 0.37** | 72 ± 9 |
| Upper-benthic Predators (FG10) | 30.3 (F)' | 165.4 ± 37.7 (24.1, A) | 101.2 ± 23.4 (28.6, D) | 62 ± 25 τ = −0.36* | 70 ± 20 τ = 0.72*** | 35 (F)' | 86.7 ± 27.5 (24.4, A) | 36.6 ± 9.9 (32.8, E) | 33 ± 30 τ = −0.33* | 70 ± 20 τ = 0.54*** |
| Benthic Herbivore/Omnivores (FG8) | 30.3 (F)' | 75.9 ± 18.5 (27, C) | 108.3 ± 15.2 (27, C) | 44 ± 31 | 71 ± 25 τ = 0.51*** | 35 (F)' | 47.8 ± 6.1 (28.5, B) | 247 ± 93.8 (30.5, D) | 27 ± 28 τ = −0.29* | 55 ± 27 τ = 0.54*** |
| Upper-benthic Planktivores (FG 2) | 30.3 (F)' | 44 ± 12.5 (27, C) | 36.3 ± 12.8 (27, C) | 30 ± 21 | 68 ± 28 τ = 0.53*** | 35 (F)' | 24.3 ± 5.5 (30.5, D) | 0.8 ± 0.5 (30.5, D) | 14 ± 16 | 65 ± 21 τ = 0.46*** |
| Upper-benthic Herbivores (FG6) | 25.3 (B) | 236.8 ± 35.2 (24.1, A) | 23.9 ± 17 (27, C) | 11 ± 16 | 64 ± 27 τ = 0.83*** | 32.8 (E) | 255.4 ± 51 (28.5, B) | 3.2 ± 1.6 (30.5, D) | 6 ± 13 | 68 ± 24 τ = 0.64*** |
| Demersal Predators (FG12) | 30.3 (F)' | 0.7 ± 0.4 (27, C) | 0.03 ± 0.04 (24.1, A) | 9 ± 19 | 56 ± 19 | 30.5 (D) | 3.4 ± 1.6 (24.4, A) | 7e-5 ± 1e-4 (32.8, E) | 1 ± 4 | 72 ± 17 τ = 0.38*** |
| Benthic Planktivores (FG4) | 28.6 (D) | 12.7 ± 3.6 (27, C) | 6e-4 ± 7e-4 (27, C) | 5 ± 14 | 62 ± 20 τ = 0.39* | 30.5 (D) | 14.6 ± 4.3 (28.5, B) | 0.02 ± 0.02 (30.5, D) | 14 ± 22 τ = 0.3* | 57 ± 20 τ = 0.62*** |
| Upper-benthic Omnivores (FG1) | 30.3 (F)' | 7.6 ± 4.7 (27, C) | 55.3 ± 31.4 (27, C) | 18 ± 21 | 47 ± 30 τ = 0.55*** | 32.7 (E) | 18.9 ± 4 (28.5, B) | 2.9 ± 1.8 (30.5, D) | 1 ± 3 | 55 ± 20 τ = 0.49** |
| Corallivores (FG16) | 28.6 (D) | 10.9 ± 3.9 (27, C) | 0 | NA | 60 ± 26 τ = 0.45* | 30.5 (D) | 7 ± 2.4 (24.4, A) | 0 | NA | 56 ± 24 τ = 0.50** |

Functional groups are ordered by decreasing species richness (Fig. 2). The tropicalisation leading edge shows the latitude and zone where a 20-fold tropical biomass decline was observed: functional groups with leading-edge latitudes that match the community-level leading edge are considered 'tropicalising as expected' (t): other functional groups are less tropicalised than expected (except Benthic Predators in Japan). Tropical-temperate functional niche overlap gives the mean ± standard deviation percentage of functional niche space occupied by tropical species that overlaps with the functional niche of temperate species across each latitudinal gradient. Tropical functional niche area gives the mean ± standard deviation percentage of tropical species' functional niche area across the latitudinal gradient relative to their functional niche area in the Tropical Coral Reef zone (A). Functional niche metrics shown here are summarised at site level and thus lower than those visualised at zone level in Fig. 4. Kendall Rank correlation (τ) between functional niche metrics and tropicalisation footprint (Fig. 4 trends) are shown if significant (<0.001 = ***, <0.01 = **, <0.05 = *). Transitional community zones are coded: A = Tropical Coral Reef, B = Cold-tropical Reef, C = Warm-transitional Reef, D = Transitional Reef, E = Subtropical Reef, F = Temperate Kelp Reef, G = Cold-transitional Reef.

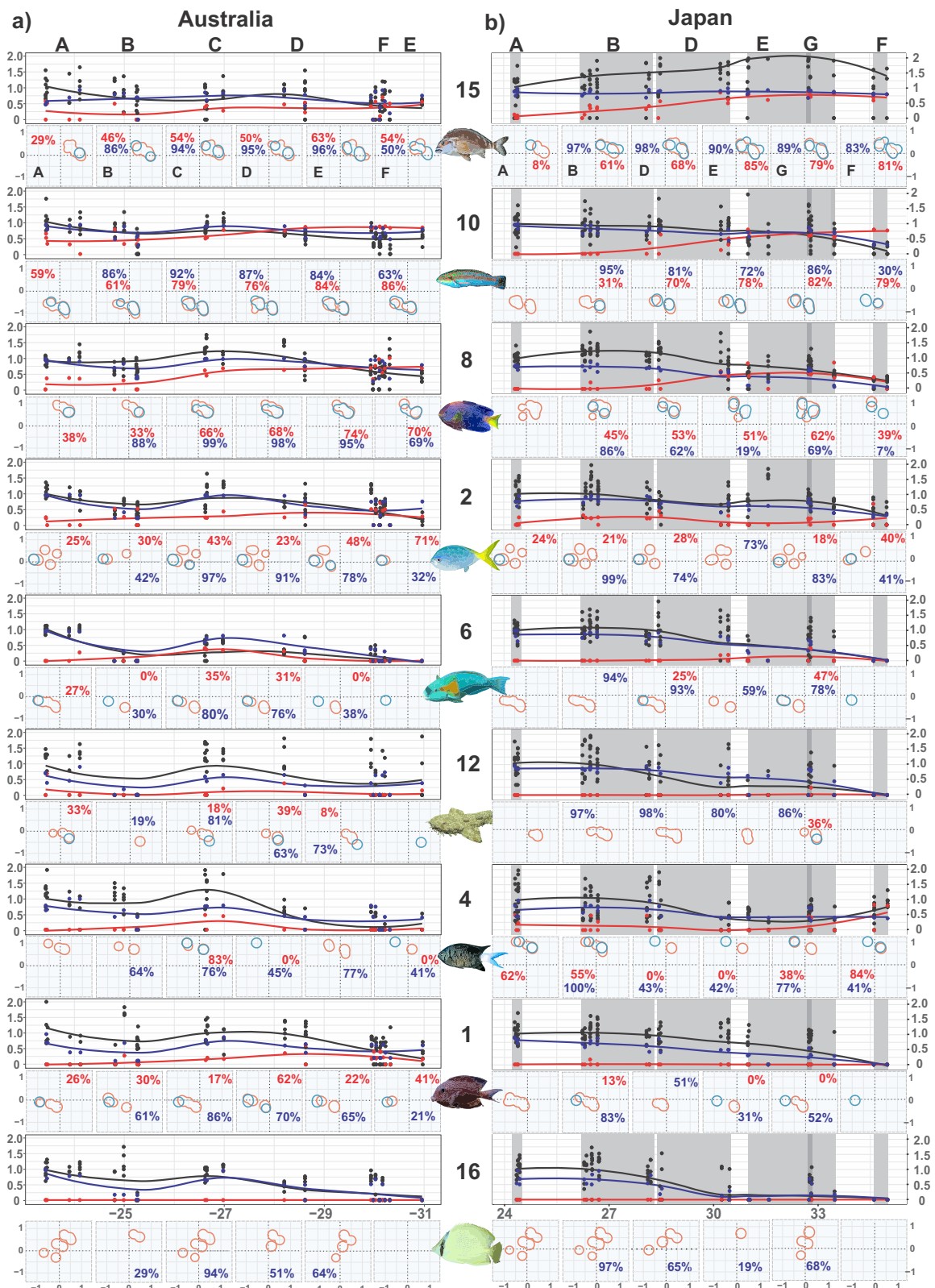

of functional overlap shows that tropical species can perform the same role as their respective temperate species (but see Benthic Planktivores FG4 for examples of functional niche partitioning; Fig. 4), and that they can invade temperate niches without displacing temperate residents or suffering competition-reduced tropicalisation.

**Occupied temperate functional niches promote tropicalisation**
Having established that thermal guild competition affects a single functional group and environmental filtering affects almost all functional groups, we evaluate which factors drive observed differences. We find that tropicalisation is higher in functional groups with occupied temperate niches, particularly when tropical species closely

**Fig. 4 | Functional niche trends over latitude.** Functional niche changes within nine functional groups (row numbers) over latitude in (**a**) Australia, and (**b**) Japan. Scatter plots show latitudinal change in tropicalisation footprint (proportion of tropical biomass relative to that of the Tropical Coral Reef zone; black), functional niche proportional overlap between tropical and temperate thermal guilds (red) and tropical functional niche area (proportion of tropical species' functional niche area relative to their functional niche area in the Tropical Coral Reef zone; blue) at site level, with trends estimated by Loess smoothers. Functional trait space plots underlying scatter plots show PCoA ordination (using first two axes) of species traits at each of the six zones (A–G). Functional niches of tropical (pink) and

temperate (light blue) thermal guilds are represented as polygons (kernel surrounding species' points in trait space) to show functional niche overlap between tropical and temperate thermal guilds (red percentage) and tropical functional niche area (blue percentage) for each zone. Transitional community zones are coded: A = Tropical Coral Reef, B = Cold-tropical Reef, C = Warm-transitional Reef, D = Transitional Reef, E = Subtropical Reef, F = Temperate Kelp Reef, G = Cold-transitional Reef. Fish icons depict species characteristic for the functional groups, created in CorelDraw 16 from original photographs by the authors. Source data are provided as a Source Data file.

match the functional niches of temperate counterparts (Table 1 and Fig. 4). We determine the drivers of functional group tropicalisation by identifying variables that explain differences in tropicalisation leading edges of functional groups, which span three zones in each region (either a high-latitude zone (F) or one of two lower latitude zones; Table 1).

The maximum tropical biomass in each functional group does not explain different leading-edge latitudes (ANOVA, $F_{(2,15)} = 0.07$, $P = 0.92$), but functional groups with greater maximum temperate biomass show higher latitude leading edges (ANOVA, $F_{(2,13)} = 9.69$, $P = 0.03$; Supplementary Fig. S6). Functional groups that exhibit greater functional overlap between tropical and temperate thermal guilds show higher latitude leading edges (ANOVA, $F_{(2,13)} = 8.6$, $P = 0.004$), but the functional area maintained by tropical species in each functional group do not explain different leading-edge latitudes (ANOVA, $F_{(2,15)} = 0.14$, $P = 0.87$). Functional groups that support greater species richness also show higher latitude leading edges than those with lower species richness (ANOVA, $F_{(2,15)} = 6.01$, $P = 0.01$). Neither realised upper thermal limit (ANOVA, $F_{(2,15)} = 1.14$, $P = 0.35$), habitat association (Chi-square test, $\chi^2_6 = 4.34$, $P = 0.65$) nor region (Chi-square test, $\chi^2_2 = 1.4$, $P = 0.5$) explain different leading-edge latitudes. Diet had no significant effect on leading-edge latitudes, with for instance herbivores and planktivores straddling high and low leading-edge latitudes (Chi-square test, $\chi^2_{10} = 10.05$, $P = 0.45$). However, predators display highest leading-edge latitudes in five of six cases (Table 1).

Our results support habitat requirements being a key determinant of functional group tropicalisation[27]. Functional groups with greater tropical-temperate functional niche overlap (e.g., FGs 15, 10, 8, 2; Table 1) have higher latitude leading edges, demonstrating the importance for tropicalising species of a pre-existing functional niche occupied by temperate residents that can be invaded. In effect, the functional role performed by these tropical species is not limited to tropical habitat and facilitates their tropicalisation. The high functional niche overlap of these functional groups is mediated by their high species richness as functional niche areas of both thermal guilds are maintained over latitude via redundancy[34]. Our a priori labelling of species-rich functional groups as generalists is supported by this representation over latitude and their greater tropicalisation[24]. By contrast, the remaining functional groups with lower latitude leading edges display low functional niche overlap and lower species richness (Table 1). This pattern is due to non-existent or very limited pre-existing temperate functional niches to invade (e.g., Corallivores FG16 and Demersal Predators FG12; Fig. 4) or smaller pre-existing temperate functional niches that tropical species struggle to match (e.g., Upper-Benthic Omnivores FG1, Benthic Planktivores FG4, Upper-benthic Herbivores FG6; Fig. 4). These functional roles are more specialised, either tied to tropical habitat (e.g., reef-building corals for Corallivores), or performed differently by temperate species. Both of these mechanisms appear to have reduced tropicalisation of these functional groups.

Further, our results show that functional niches at a higher latitude with high temperate biomass can better accommodate tropicalisation than those niches with low temperate biomass. The more tropicalised generalist functional groups mentioned above have

relatively high temperate biomass that is comparable to their tropical biomass despite much lower species richness (Table 1 and Fig. 2). The remaining, less tropicalised, functional groups have low temperate biomass, or for Upper-benthic Herbivores FG6, temperate biomass that is an order of magnitude lower than tropical biomass, both describing functional niches that have less capacity to support biomass at higher latitudes than in the tropics.

## Discussion

We show tropicalisation footprints in functional groups of reef fishes are mediated by environmental filtering and pre-existing temperate functional niches, rather than competition between tropical and temperate thermal guilds. Thus, Darwin's naturalisation hypothesis is not supported, as only a single group (Upper-benthic Predators FG10) demonstrates that increased sharing of niche space with temperate counterparts reduces tropicalisation. However, such competition does not prevent the tropicalisation leading edges from reaching high latitudes in both regions. Instead, our finding that tropical species in functional groups that better match the functional niche (overlap) of their temperate counterparts show greater tropicalisation aligns with Darwin's pre-adaptation hypothesis, supporting the notion that tropicalising species obtain an advantage if they occupy similar functional space as resident temperate species, pre-adapting them to their new environment[31]. Our functional group approach captures pre-adaptation of tropical species to perform the function of temperate counterparts via traits. Indeed, lowest tropicalisation is observed in functional groups where temperate niches were limited or hard to match. In the more tropicalised functional groups, where tropical and temperate species share functional niches, co-existence indicates resources are not limited for tropical species at higher latitudes[30]. Two non-mutually exclusive mechanisms could explain such relaxation of competitive restraints. Firstly, the recipient transitional-temperate communities may not have reached carrying capacity, for example, phase shifts (kelp to turf) may facilitate microhabitat and resource availability[30]. Alternatively, tropical species may have enough plasticity to exploit different micro-niches to temperate residents, even within the same functional group[46]. Our results support both mechanisms, as we show that, based on the traits used in our study, more tropical species are packed into each functional niche than temperate species (i.e., have higher functional redundancy), indicting higher competition pressure on tropical coral reefs may predispose fish to exploit smaller micro-niches.

The broad scope of our study warrants a cautious interpretation of findings relative to those of site or species-specific studies. Our large functional groups undoubtedly mask finer scale competition that could be explored further by calculating functional overlap weighted by group biomass (Supplementary Table S2 and Supplementary Fig. S7) or at species level[29,46]. The identity of functional groups also changes over latitude as their species turnover through environmental filtering[18]. In species-rich functional groups, labelled as representing generalist ecosystem functions (e.g., Benthic Predators FG15), filling of functional space is maintained, while in more specialist functional groups (e.g., Corallivores FG16) filling of functional space is reduced. Although our labelling of generalist/specialist functional groups

supports greater tropicalisation in generalists, an examination of their constituent species would likely reveal both generalist and specialist species within each functional group, with the identity of functional groups dominated by generalists at range leading edges[12,24]. Furthermore, the latitudinal gradients of our study potentially represent multiple phases of invasion (introduction, establishment, self-sustaining populations), within which processes such as competition could be interpreted differently. For example, the naturalisation hypothesis suggests that the niche partitioning observed in Benthic Planktivores FG4 should represent 'niche opportunities', promoting tropicalisation[29]. However, we find the opposite, which suggests we may be observing the consequences of competition that are associated with later phases of invasion.

Despite these caveats, broad-scale community change studies provide the context to reveal big-picture insights[14]. We find strong site effects on tropicalisation, demonstrating how habitat can offset latitude. For example, the Cold-tropical Reef zone in Australia shows comparable tropicalisation footprints to the high-latitude Temperate Kelp Reef zone, despite being adjacent to the Great Barrier Reef. In both regions, the clustering of fish biomass into zones identifies different communities that overlap in latitude (Subtropical Reef and Temperate Kelp Reef zones in Australia; Cold-transitional Reef and Subtropical Reef, Japan). In both cases, tropicalisation is higher in the Subtropical Reef zone, which is better positioned to intercept western boundary currents (East Australian Current or the Kuroshio Current, respectively) that facilitate tropicalisation via transport of warm tropical water and larvae[9]. By the same margin, sites that do not frequently receive tropical water from currents could act as refugia for temperate species[47,48]. Our findings suggest that the interaction between site-specific habitat and geographically dynamic boundary currents creates the potential for a diverse patchwork of differently tropicalised sites[11,48], rather than a single gradient of tropicalisation following latitude.

Globally, ongoing species redistribution and the resulting community transformations in response to changing climate have been altering the character and ecosystem services of tropical, subtropical, and temperate reefs[4–6], but a community perspective of occurrence and consequences of such tropicalisation has been lacking[14]. Our study highlights ubiquitous tropicalisation footprints in nine major functional groups of reef fishes, but differential tropicalisation pressure from these groups acting upon communities across the tropical-temperate gradient. For example, Transitional Reef communities sustain higher pressure from tropical benthic herbivory/omnivory and upper-benthic omnivory, and lower pressure from corallivory and benthic planktivory relative to Tropical Coral Reef communities. The demonstrated asynchronous invasion of higher latitude functional niches by tropical species helps unravel how relationships between functional groups underpin tropicalisation phase shifts[5], while our findings that reef fish tropicalisation is mostly shaped by tropical species' ability to persist in high-latitude environmental conditions not biotic interactions (such as competition) explain the mechanisms behind the asynchronous invasion. Together our results illustrate the importance of interacting ecosystem functions and environmental drivers in transforming coastal reef communities and their implications for resilience of temperate ecosystem function under increasing tropicalisation.

## Methods

### Fish surveys and community clustering

Our research complies with all relevant ethical regulations, with permits and approvals obtained from New South Wales and Queensland departments of Primary Industry, Great Barrier Reef Marine Park Authority and University of Queensland. Reef fish abundance and biomass of all non-cryptic species associated with coral communities were recorded at 54 sites across a latitudinal gradient from 23° to 34° N and S in Japan (29 sites) and Australia (25 sites) (Fig. 1 and Supplementary Fig. S1), encompassing the poleward range edge of tropical species and cooler-water sites experiencing tropicalisation by range-shifting tropical species. At each site, numbers and sizes (to the nearest cm) of all non-cryptic fish were recorded along 3–5 replicate belt transects of 25 m or 50 m length and 5 m width in coral communities (see Supplementary Data 3 for a species list). Site depths were standardised at 8–10 m, with some temperate coral communities surveyed shallower (between 3 and 6 m, two sites in Australia and one site in Japan). The biomass $B$ of each recorded fish was calculated from tail length with $B = a*fishlength^b$ (a, b from fishbase;[49]). Surveys were undertaken in Boreal summers (Jun–Jul) of 2015 or 2016 in Japan, and in Austral winter (Aug–Oct) of 2010, 2011, 2012, 2016, 2017 and 2018 in Australia. Sites in Japan were surveyed in 1 year only, but those in Australia were surveyed between 1 and 6 years (repeat visits). To account for differences between sites due to different transect lengths, number of transects, and the number of repeat years surveyed, we standardised species biomass per unit area of transect $(g/m^2)$ and used transect as our sampling unit for analyses. Using R package vegan (version 5.2.7), log-transformed standardised biomass was input into a site by species matrix, and calculated Bray Curtis distances were entered into 'average' method hierarchical clustering to group sites into transition zones across each regional latitudinal gradient (Supplementary Fig. S1), and explore the drivers of community turnover (Supplementary Results).

### Fish traits and FG identification

We determined the broad fundamental functional niches within transitional fish communities by categorising fish species into functional groups (Supplementary Fig. S1, Supplementary Data 1, and sensu[16]). Species from both regions were combined into a meta-community to identify functional groups relevant to both regions and reflect similar pressures shaping community-wide fundamental functional niche space. Additional Australian surveys in Austral summer, from the same sites and within the same year range, were used to augment the species list (+64 species), and help build a complete community inclusive of both summer and winter seasons. Using R package cluster (version 2.1.2), Gower distances between species were calculated from the five traits (body size and depth range log-transformed, aggregation treated as ordinal) and a dendrogram created using hierarchical clustering with the 'average' method that best represented the original distances[50]. There were few missing values in the trait database, only 18 species had an NA for depth range and one species with an NA for diet. The optimal number of clusters (functional groups) was selected from the dendrogram based on inspection of the average silhouette width, the Jaccard similarity index and Rand matching index (R package clue, version 2.1.0) while performing a 1000-iteration bootstrap of the original data[51] (to incorporate cluster stability). That is, during each iteration 5% of the data was randomly omitted and the distance matrix recalculated (with scaling of the original matrix), followed by calculation of each index over two to 30 clusters (Supplementary Figs. S3 and S4).

### Analyses of tropicalisation

To compare functional group tropicalisation trends, we compare 'tropicalisation footprints' between functional groups in communities across latitude. For each functional group and the community as a whole, we total tropical biomass at the transect level and calculate proportional change between each transect and the average total biomass of transects in the lowest latitude zone, that is:

$$BT_{ij} = \frac{b_{ij}}{\bar{b}_i} \tag{1}$$

where the tropicalisation footprint of FG $i$ at transect $j$ outside the Tropical Coral Reef zone ($BT_{ij}$) is equal to the total proportion of biomass $b$ of species within FG $i$ recorded at high-latitude transect $j$, relative to the mean biomass ($\bar{b}$) of species in functional group $i$ recorded across all transects within the Tropical Coral Reef Zone.

The community-wide biomass change provides a region-specific baseline for the proportion of tropical biomass expected at increasing latitudes, i.e., the community-wide rate of tropicalisation. Within zones of increasing latitude, we test whether the tropical biomass change of functional groups differed significantly from that of the whole community. We create linear mixed effects models for each zone (excluding the Tropical Coral Reef zone) with proportional change in biomass fitted against functional group, and including site as a random effect to account for similarities between transects from the same site with R package nlme (version 3.1-157). We apply a fourth-root transformation to biomass proportions to reduce the effect of large biomass change at some transects while respecting zeros arising from functional group absence at other transects.

We summarise the tropicalisation of each functional group using the poleward tropicalisation leading edge, defined as the latitude of the zone in which each functional group's tropicalisation footprint first showed a 20-fold decline in tropical species biomass. A 20-fold decline was chosen because it represents the decline in community-level biomass between the Tropical Coral Reef zone and the Temperate Kelp zone. If a functional group's tropicalisation leading-edge occurred in the same zone as the 20-fold decline in the community tropicalisation footprint (community-level tropicalisation leading edge) then we class it as 'tropicalising as expected'. If, instead, it occurred in a lower latitude zone than the community-level leading edge (two zones were observed in each region), then it is classed as 'tropicalising below community expectation' or 'tropicalising well below community expectation', respectively.

## Analyses of functional niche

To explore the area and overlap of realised functional niches, we construct a multidimensional trait space. We apply Principal Coordinates Analysis (PCoA) ordination of the Gower trait distance matrix (with Cailliez correction) and extract the first four PCoA axes (explaining 19%, 16%, 10%, and 7% of the variance, respectively) using R package Ade4 (version 1.7-17). We create six two-dimensional representations of functional space through the pairwise combination of the four PCoA axes. Functional niche metrics are calculated within each functional space and then averaged together using the variance explained by each space's PCoA axes pair as a weight. As all species from both regions are used for functional space construction, it allows comparison of functional niches between thermal guilds, functional groups and regions. The functional niche area of each thermal guild within each functional group is delineated from the 99% Utilisation Distribution of a kernel surrounding the species (points in trait space) occurring at each site, using R package adehabitatHR (version 0.4.19). This approach is analogous to the measure of functional richness using convex hulls, but the kernel method does not assume a functional group can exploit all of a niche area delineated by its most functionally distinct members, which is appropriate when dealing with our broad functional groups. Functional niches for both thermal guilds within each functional group are created at the site level for analyses and at the zone level for visualisation. We calculate environmental filtering from the change in tropical functional niche area between the lowest latitude zone and that of each site over the latitudinal gradient. To measure thermal guild competition, we calculate functional niche overlap at each site, expressed as the proportion of the tropical functional niche area covered by the temperate functional niche area. If zero functional niche overlap was observed in any of the six representations of functional space then average overlap for the site was also forced to zero.

We test whether environmental filtering and competition between tropical and temperate thermal guilds are related to tropicalisation footprints within functional groups with non-parametric Kendall rank correlation. Correlation analysis is performed at the site level across each latitudinal gradient, using either proportional change of tropical functional niche area or proportional functional niche overlap of thermal guilds and site-level tropicalisation footprints estimated by linear mixed models. Values where the tropicalisation footprint is zero (forcing both functional metrics to zero also) are removed to prevent inflated correlation.

To explain differences in functional group tropicalisation leading edges, we test the significance of several functional and environmental drivers. Using each driver in turn as the response variable, we fit models with tropicalisation leading-edge class (tropicalising as expected, below expectation, or well below expectation) as a categorical explanatory variable, combining data from both regions. To test whether differences in tropicalisation leading edge are explained by maximum biomass of thermal guilds, we estimate the mean tropical and temperate biomass of each functional group within each zone, and use the maximum value in generalised linear models (GLM) with gamma error structure. To test whether differences in tropicalisation leading edge is explained by tropical functional niche area or tropical-temperate functional niche overlap we average both metrics at the site level across each latitudinal gradient and fit GLMs with quasi-binomial error structure. We test whether differences in tropicalisation leading edge is explained by species richness with a GLM with quasi-Poisson error structure and whether differences in tropicalisation leading edge is explained by realised upper thermal limit with a linear model (results in Supplementary Fig. S6). We test for association between tropicalisation leading-edge class and categorical variables: diet, position and region using chi-squared tests. All manuscript figures were prepared with R package ggplot2 (version 3.3.3) and Corel-Draw (versions 16 and 20).

## Reporting summary

Further information on research design is available in the Nature Portfolio Reporting Summary linked to this article.

## Data availability

Data for the occurrence in each region, trait values and functional groups of all fish species analysed in this paper are available in Supplementary Data 1. Fish abundance, biomass and raw survey data are subject to controlled access to protect the novelty of collaborative papers still in preparation but are available from Maria Beger (m.beger@leeds.ac.uk) upon request. Source data are provided with this paper.

## Code availability

The R code written to perform all analyses is available at https://github.com/lark-gorilla/coral_fish/blob/master/code.R.

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

## Acknowledgements

The authors acknowledge funding from the Australian Research Council Centre of Excellence for Environmental Decisions (CE110001014), an EU Marie Skłodowska-Curie Fellowship (TRIM-DLV-747102) and a NERC grant (NE/S006931/1) to M.B., the Japanese Society for the Promotion of Science (JSPS) 'Zuno-Junkan' grant entitled "Studies on origin and maintenance of marine biodiversity and systematic conservation planning" to J.D.R., the Australian Research Council Centre of Excellence for Coral Reef Studies (CE140100020) to J.M.P. and others, and a Chancellor's Postdoctoral Research Fellowship from the University of Technology Sydney and a University of Sydney Fellowship to B.S.

## Author contributions

M.G.R.M. and M.B. conceived the study, analysed the data, and wrote the paper. M.B., J.D.R., B.S., K.C., J.M.P., and M.O. collected the data and edited the paper.

## Competing interests

The authors declare no competing interests.
