## [Peer Review File · Nature Communications]

Temperate functional niche availability not resident-invader competition shapes tropicalisation in reef fishesREVIEWER COMMENTS

Reviewer #1 (Remarks to the Author):

Summary and main comments

Miller and colleagues assess the footprint of tropicalisation onto temperate reefs across nine functional groups of reef fishes in eastern Australia and Japan. The authors find that habitat availability is a strong determinant of the poleward range-shifts in reef fishes and that tropical and temperate species of the same functional group tend to share functional niche space, rather than displace one another.

I found the paper interesting to read and well written. I liked the approach of including two regions to provide comparison and generality to the findings and I appreciate the large body of work that has gone into this study to analyse the full fish community across a large geographical area.

The main issue I had with the paper was the lack of detail and clarity on the methods. In particular, how traits were assigned to species and what the definition of each of the traits was. For example, I could not find an explanation of the difference between an 'upper-benthic' and 'benthic' species? This made the results confusing to me because I couldn't interpret the difference between functional groups like 'upper-benthic herbivores' and 'benthic herbivores' which demonstrated quite different results. A summary table with the definition of each category would be helpful. Also a species list or list of common species within each functional group in the SM would be helpful to visualise the difference between groups.

Other trait categories were similarly vague and require greater explanation in the methods. For example depth range. From the SM, I interpret these to be 0-30m, 31-100 and >100m. How are these deeper categories relevant to the fish you saw on your transects which were only conducted in sites 0-10m? In addition to a summary table, an explanation of whether traits categories were assigned empirically from in situ observations, expert knowledge or published sources would be helpful.

An additional concern I had was how thermal affinities were attributed and calculated. From the SM, the thermal midpoint within functional groups mostly sit above 31°C which seems incredibly high for species from these two regions – even tropical species (See Stuart-Smith et al, 2015, 2017). I suggest the calculation of these midpoints be checked or alternative products be used to estimate the thermal range of species. It was also not clear to me how tropical and temperate guilds of species were determined? For example, was this done empirically based on thermal midpoints for each species, or other means? My lack of confidence in the midpoints presented in Fig. S4 makes me question the affinities assigned to other species. Also, and this is aesthetic, but in the fish icons in Fig. 2, you have a temperate species for benthic predator and the remainder tropical species. I appreciate they are 'typical species' from the functional group – but given the uncertainty mentioned above and the fact that the results are primarily focussed on the contribution of tropical species to fish assemblages, it would make more sense to include a tropical species there too.

Greater clarification on the methods would help resolve all of my concerns and I otherwise think the paper is very relevant and of strong interest to the field. Below are some additional minor comments with line numbers, some of which reflect my main comments described above.

Minor comments

L87-90: Sentence difficult to understand. Consider rewording.

L131: How were the 5 traits categorised for each species? How was habitat association defined for example? I cant find this explanation in methods. In the SM I interpret the habitat association categories as demersal, benthic, upper benthic, subbenthic – but it is still unclear to me what the definition of these categories is or how species were attributed to a category.

L131: How was depth range classified? From the SM, I interpret these to be 0-30m, 31-100 and >100m. How is this relevant to the fish you saw on your transects which were only conducted in sites 0-30m?

L137-139: More detail is required on how thermal midpoints were calculated for species

here. Fig S4 indicates that thermal midpoints are $>31^{\circ}\text{C}$ within all of the functional groups. That temperature seems very high as a midpoint – particularly for species in Australia.

137-138: What thermal affinities were assigned to each species to determine whether they were tropical or temperate? Did you calculate thermal midpoints for each species and empirically determine affinity? It would help to see this information, or at least an explanation in the methods.

L142: I find myself constantly questioning where these classifications come from. Could you please clarify what the different habitat affinity and dietary classifications are how they were assigned to species?

L169: Again, what is the difference between a benthic herbivore and an upper-benthic herbivore: Please define habitat categories. A table of species list and functional group categories assigned to species would be helpful for interpretation.

L194-195: But didn't your surveys only record fishes? How do invertebrates factor into this result?

L253-256: habitat is attributed as the most important driver of functional group tropicalisation, but nowhere are these habitat classifications defined, and it isn't explained how species were placed into one or another. From what I infer, these classifications are not related to biogenic habitat association per se – but...place in the water column? No mention of habitat being recorded is mentioned in the methods that I can find – so were these taken from a published source? Given so much is riding on these habitat associations, more clarity is needed around what they actually mean.

L362: What is meant by 'tail length' total length or fork length?

L363-364: How do the differences in season influence the fish assemblages observed at each location? Particularly toward the higher latitudes, there can be strong differences in the fish activity between summer and winter, influencing which species are picked up on transects and their estimated abundance.

L364: How many sites were surveyed in each country? The map in Fig. 1 suggests 3-5 sites per climate zone, but methods currently only mention 3-5 replicate transects within a site. Consider elaborating on the survey design.

L380: I know you specify the five traits in the results, but it would help to repeat them here .

L390-400: How did you delineate between tropical and temperate species? (e.g. Measure of thermal affinity, expert knowledge)

L393: Word missing in "we total tropical biomass"

SM Fig S2 caption: – There is no fish icon shown in figure.

SM Fig S4: What temperature data was used to calculate the thermal midpoints?

Thermal midpoints above 31°C for all of the functional groups seems highly unrealistic – particularly for species in these two regions. In Stuart Smith et al. 2015 for instance, almost all tropical species had thermal midpoints below 30°C .

Reviewer #2 (Remarks to the Author):

This article investigates the potential effect of topicalization of Western Pacific ecosystems using a space-for-time approach.

Biomass of >500 fish species were monitored along 2 gradients of temperature with 6 zones each (southward of Japan and northward of Australia).

The fish species were clustered into functional groups and trait overlap between temperate and tropical fishes were computed for each group.

I have 5 main comments about the methods used in this study:

1) Surveys were undertaken in the warmest season for the Japanese zones but in the coldest one for Australian zones. Could the difference in sampling period have influenced the detection (and/or biomass estimates) of tropical species in the temperate zones of Australia? Based on statement line 379 it seems that many species do occur only during Austral summer in Australia, but there is no information about their biomass

during these months, hence their potential competition with temperate species and effects on ecosystems.

2) Clustering of species is done based on a dendrogram computed on Gower distance between species (described with 5 traits). Number of groups was selected based on comparisons between several cut-offs. However, there is no information about the quality of the dendrogram while previous studies reported dendrograms often bias the distance between some species (Maire et al 2015 ; DOI: 10.1111/geb.12299). What is the accuracy of the dendrogram computed here? As overlap analyzes are based on a multidimensional space built using PCoA why not clustering species using the k-means algorithm applied to Euclidean distances in this space ?

3) 19 groups were identified on the dendrogram but only 9 of them were kept for overlap analyzes. Even if those 9 groups gather 85% of the species, some excluded groups could have few species but a high biomass. In addition, some excluded groups (e.g. FG 5 with 24 species) have more species than selected groups (max = 20 species). It would be of interest to test for the overlap between tropical and temperate species in all groups. Related to this point, number of species per group differ between Figure 2 and Figure S2 (e.g. 133 vs 141 species for FG15). Which numbers are correct?

4) Overlap between tropical and temperate species is computed in a 2D space built on trait-based distance. However, such low-dimensionality spaces are often underestimating distance between species (Maire et al 2015). Please provide quality metrics of the PCoA and consider accounting for more PC axes to compute overlap.

5) Overlap is computed based on a metric accounting only for presence of species. Therefore, it does not discriminate cases where both temperate and tropical species have high biomass, from those where one of the 2 species is rare. It would thus be complementary to compute a biomass-weighted similarity metric (e.g. Mean Nearest Neighbor Distance) to test whether increasing overlap based on occurrence is paired with increasing biomass of the most closely related species.

Minor comments:

- Line 51: this first sentence would be clearer if the driver of species redistribution was named here not in following sentence.
- Line 63: this sentence about functional groups could be moved after the trait-based approach presented in lines 64-67
- Line 69: it is unclear why functional niche is defined for species with similar traits, since a trait-space could only be built if all species were described with the same traits. Do you mean functional groups gathering species with same trait values?
- Lines 87:90: add names of first author for ref 28, 29 and 30
- Lines 143-144: specialist vs generalist generally refer to species ecology (or diet) based on the range of suitable environments (or preys eaten). Here definition is rather similar to the redundancy concept since group with many species are likely to have more species with similar trait values (although it needs to be confirmed using distance-based metrics)
- Line 149: "generally shows" is vague. How many groups have such a pattern and what is the average slope?
- Line 187-196: this section about implications of findings could be moved to discussion, and adding names of fish species with these features would be more informative.
- Line 376: given the large spatial extent of the study these are not meta-communities, rather meta-ecosystems

Response to reviewer comments

Temperate functional niche availability not resident-invader competition shapes tropicalisation in reef fishes

Mark Miller*, James D. Reimer, Brigitte Sommer, Katie M. Cook, John M. Pandolfi, Masami Obuchi, Maria Beger*

REVIEWER COMMENTS

Reviewer #1 (Remarks to the Author):

Miller and colleagues assess the footprint of tropicalisation onto temperate reefs across nine functional groups of reef fishes in eastern Australia and Japan. The authors find that habitat availability is a strong determinant of the poleward range-shifts in reef fishes and that tropical and temperate species of the same functional group tend to share functional niche space, rather than displace one another. I found the paper interesting to read and well written. I liked the approach of including two regions to provide comparison and generality to the findings and I appreciate the large body of work that has gone into this study to analyse the full fish community across a large geographical area.

The main issue I had with the paper was the lack of detail and clarity on the methods. In particular, how traits were assigned to species and what the definition of each of the traits was. For example, I could not find an explanation of the difference between an 'upper-benthic' and 'benthic' species? This made the results confusing to me because I couldn't interpret the difference between functional groups like 'upper-benthic herbivores' and 'benthic herbivores' which demonstrated quite different results. A summary table with the definition of each category would be helpful. Also a species list or list of common species within each functional group in the SM would be helpful to visualise the difference between groups. Other trait categories were similarly vague and require greater explanation in the methods. For example depth range. From the SM, I interpret these to be 0-30m, 31-100 and >100m. How are these deeper categories relevant to the fish you saw on your transects which were only conducted in sites 0-10m? In addition to a summary table, an explanation of whether traits categories were assigned empirically from in situ observations, expert knowledge or published sources would be helpful.

Response 1: We thank the reviewer for highlighting these method issues. We have now provided several new items to clarify the methods:

- Table for trait definitions and data provided in the Supplement (Table S1);
- Species names for each functional group provided in the Supplement (Table S3); and
- Trait and species data provided in the Supplement (Table S3).

We clarify that the depth range values are continuous, there are no categories. This is now clear from the trait definition table.

An additional concern I had was how thermal affinities were attributed and calculated. From the SM, the thermal midpoint within functional groups mostly sit above 31°C which seems incredibly high for species from these two regions – even tropical species (See Stuart-Smith et al, 2015, 2017). I suggest the calculation of these midpoints be checked or alternative products be used to estimate the thermal range of species. It was also not clear to me how tropical and temperate guilds of species were determined? For example, was this done empirically based on thermal midpoints for each species, or other means? My lack of confidence in the midpoints presented in Fig. S4 makes me question the affinities assigned to other species.

Response 2: We thank the reviewer for spotting the high thermal midpoint values. Stuart-Smith et al. (2015) calculate the fifth and 95th percentiles of the temperature distribution occupied by each species, and subsequently calculate the 'thermal midpoint' between these as a measure of central

tendency. When provided with the thermal midpoint dataset we were advised to use the 95th percentile of species' thermal distributions as a measure of contemporary realized upper thermal limits, because, as Stuart-Smith et al. (2015) state: *"Realized upper limits will be lower than fundamental limits based on physiological tolerances, but arguably better reflect real-world limits, where species not only need to survive physiologically, but also persist in a competitive and predatory environment"*. Our use of the 95th percentile of species' thermal distributions explains the high 'midpoint' values observed by the reviewer. To signpost the use of the 95th percentile, we have adjusted the main manuscript text on lines 138, 248 and 464, and the supplementary material: method text on lines 42-49, results on line 98-100, and updated Figure S4 and its caption.

Regarding assigning species to tropical and temperate guilds, we based decisions on data obtained from FishBase (www.fishbase.org) and expert opinion by author MB (which is informed by fish observations on tropical, subtropical and temperate reefs spanning over 25 years of experience). Fig S4 shows realized upper thermal limits for the species where these data were available (from Rick Stuart-Smith, University of Tasmania). The uncertainties about these realized upper thermal limits are exactly why we used expert opinion to assign the thermal guilds, but we wanted to also show the variance about these estimates.

Also, and this is aesthetic, but in the fish icons in Fig. 2, you have a temperate species for benthic predator and the remainder tropical species. I appreciate they are 'typical species' from the functional group – but given the uncertainty mentioned above and the fact that the results are primarily focussed on the contribution of tropical species to fish assemblages, it would make more sense to include a tropical species there too.

Response 3: We reassigned the "typical" species for most of the groups to ensure that we comply with copy right issues on the images, and in this process, we now have a more even mix of temperate and tropical species.

Greater clarification on the methods would help resolve all of my concerns and I otherwise think the paper is very relevant and of strong interest to the field.

Below are some additional minor comments with line numbers, some of which reflect my main comments described above.

Response 4: We thank the reviewer for pointing out these minor issues, we have addressed them all, as per short indication below:

Minor comments

L87-90: Sentence difficult to understand. Consider rewording. **Reworded.**

L131: How were the 5 traits categorised for each species? How was habitat association defined for example? I cant find this explanation in methods. In the SM I interpret the habitat association categories as demersal, benthic, upper benthic, subbenthic – but it is still unclear to me what the definition of these categories is or how species were attributed to a category. **Supplementary trait definition Table S1.**

L131: How was depth range classified? From the SM, I interpret these to be 0-30m, 31-100 and >100m. How is this relevant to the fish you saw on your transects which were only conducted in sites 0-30m? **This trait is a continuous data set and not classified for the main PCOA trait space analyses in the paper; however, it is classified (as above) to create wordclouds to help visualise functional groups. This has been better signposted in L25 of the supplementary material, and underscored in**

the newly added supplementary trait definition Table S1.

L137-139: More detail is required on how thermal midpoints were calculated for species here. Fig S4 indicates that thermal midpoints are $>31^{\circ}\text{C}$ within all of the functional groups. That temperature seems very high as a midpoint – particularly for species in Australia. See response 2. Details added to Supplement/ figure caption.

137-138: What thermal affinities were assigned to each species to determine whether they were tropical or temperate? Did you calculate thermal midpoints for each species and empirically determine affinity? It would help to see this information, or at least an explanation in the methods. See response 2.

L142: I find myself constantly questioning where these classifications come from. Could you please clarify what the different habitat affinity and dietary classifications are how they were assigned to species? Supplementary trait definition Table S1. Assignment in Supplementary Table S3.

L169: Again, what is the difference between a benthic herbivore and an upper-benthic herbivore: Please define habitat categories. A table of species list and functional group categories assigned to species would be helpful for interpretation. Supplementary trait definition Table S1, S3.

L194-195: But didn't your surveys only record fishes? How do invertebrates factor into this result? We point to invertebrate herbivory here because we find that one fish functional group has a tropicalisation footprint below the community average, and tropicalisation driven by herbivory likely is not only attributable to fish herbivores, but also invertebrates. The new text reads as follows: *"However, our results support a careful choice of herbivore species with which to infer community tropicalisation; for example, Upper-benthic Herbivores in Australia were consistently less tropicalised than the community. Given that herbivory facilitates tropicalisation in eastern Australia (Verges et al. 2016), these processes likely illustrate the importance of invertebrate herbivory in concert with fish grazing (Ling et al. 2009)."* (Lines 192-197).

L253-256: habitat is attributed as the most important driver of functional group tropicalisation, but nowhere are these habitat classifications defined, and it isn't explained how species were placed into one or another. From what I infer, these classifications are not related to biogenic habitat association per se – but...place in the water column? No mention of habitat being recorded is mentioned in the methods that I can find – so were these taken from a published source? Given so much is riding on these habitat associations, more clarity is needed around what they actually mean. Thank you for finding this inconsistency in the text. We did not include habitat classifications, as all our data comes from shallow reefs and should be seen as equivalent across the climate zones. On (now) line 254, we changed the text to clarify what we mean, now reading: *"Our results show that existing features, such as suitable habitat or existing communities, shape functional group tropicalisation"*.

L362: What is meant by 'tail length' total length or fork length? Corrected to total length.

L363-364: How do the differences in season influence the fish assemblages observed at each location? Particularly toward the higher latitudes, there can be strong differences in the fish activity between summer and winter, influencing which species are picked up on transects and their estimated abundance.

Response 5: Quantifying the temporal differences in fish activity and assemblages recorded on transects is beyond the scope of this study. However, we are reasonably confident that within the confines of our survey methods, the spatial patterns in fish assemblages dominate over temporal

ones. For example, at the Solitary Islands Marine Park (containing several of our Australian high latitude sites), the temporal variability in fish assemblages is less than the spatial variability (Malcolm et al 2007; Malcolm and Ferrari 2019). Similarly, no seasonal signal was observed in fish and benthic assemblages at Julian Rocks (also a high latitude site in Eastern Australia, Sommer et al 2014). Similarly, in Japan, seasonal differences were not detectable for rockpool fish communities (Murase 2013), though seagrass associated fish communities in the Seto Sea displayed seasonal patterns with changing shoot density of seagrass (Mohri et al 2013). Whilst it is unclear how fish communities that associate with coral communities change seasonally, we expect them to respond similarly to those associated with other rocky substrates. Also, in Japan all observations were done in summer, and seasonal differences do not influence our dataset.

References:

- Malcolm, H. A., W. Gladstone, S. Lindfield, J. Wraith, and T. P. Lynch. 2007. Spatial and temporal variation in reef fish assemblages of marine parks in New South Wales, Australia - baited video observations. *Marine Ecology-Progress Series* **350**:277-290.
- Malcolm, H. A., and R. Ferrari. 2019. Strong fish assemblage patterns persist over sixteen years in a warming marine park, even with tropical shifts. *Biological Conservation* **232**:152-163.
- Sommer, B., M. Beger, and J. M. Pandolfi. 2014. Seasonal and temporal changes in reef fauna at Julian Rocks 2010 to 2013. The University of Queensland, Brisbane, Australia.
- Murase, A. 2013. Community structure and short temporal stability of a rockpool fish assemblage at Yaku-shima Island, southern Japan, northwestern Pacific. *Ichthyological Research* **60**:312-326.
- Mohri, K., Y. Kamimura, K.-i. Mizuno, H. Kinoshita, S.-i. Toshito, and J. Shoji. 2013. Seasonal Changes in the Fish Assemblage in a Seagrass Bed in the Central Seto Inland Sea. *Aquaculture Science* **61**:215-220.

L364: How many sites were surveyed in each country? The map in Fig. 1 suggests 3-5 sites per climate zone, but methods currently only mention 3-5 replicate transects within a site. Consider elaborating on the survey design. We included that there are a total of 54 study sites (29 in Japan, 25 in Australia), each of which has 3-5 replicate transects, line 358.

L380: I know you specify the five traits in the results, but it would help to repeat them here. Done. Line 377.

L390-400: How did you delineate between tropical and temperate species? (e.g. Measure of thermal affinity, expert knowledge) See Response 2.

L393: Word missing in “we total tropical biomass” Done, added “add”, line 395.

SM Fig S2 caption: – There is no fish icon shown in figure. The sentence referring to fish icons was removed.

SM Fig S4: What temperature data was used to calculate the thermal midpoints? Thermal midpoints above 31°C for all of the functional groups seems highly unrealistic – particularly for species in these two regions. In Stuart Smith et al. 2015 for instance, almost all tropical species had thermal midpoints below 30°C. See Response 2. The data shows realised upper thermal limits, this was amended.

Reviewer #2 (Remarks to the Author):

This article investigates the potential effect of topicalization of Western Pacific ecosystems using a space-for-time approach. Biomass of >500 fish species were monitored along 2 gradients of temperature with 6 zones each (southward of Japan and northward of Australia). The fish species were clustered into functional groups and trait overlap between temperate and tropical fishes were computed for each group.

I have 5 main comments about the methods used in this study:

1) Surveys were undertaken in the warmest season for the Japanese zones but in the coldest one for Australian zones. Could the difference in sampling period have influenced the detection (and/or biomass estimates) of tropical species in the temperate zones of Australia? Based on statement line 379 it seems that many species do occur only during Austral summer in Australia, but there is no information about their biomass during these months, hence their potential competition with temperate species and effects on ecosystems.

Response 6: See response 5 about seasonal changes in fish communities that is not significant in previous studies in Australia's high latitudes. Further, if we posit (inherent in the reviewer comment) that more tropical species might be present in summer, then our tropicalisation footprints are valid and provide a conservative estimate of how influential tropical species are. Our paper does not aim to compare tropicalisation between Australia and Japan- we simply show equivalent responses in both locations. There were 64 extra species only observed in Austral summer in Australia, as highlighted in Supplementary Figure Caption Fig S2. See Response 8 below for why we felt it appropriate to include in functional group creation even without corresponding biomass in winter – basically even though the extra 64 species are summer migrants they still shape functional groups. To clarify these issues, we have explained the Australian results in the context of winter detection/competition on lines 379-382.

2) Clustering of species is done based on a dendrogram computed on Gower distance between species (described with 5 traits). Number of groups was selected based on comparisons between several cut-offs. However, there is no information about the quality of the dendrogram while previous studies reported dendrograms often bias the distance between some species (Maire et al 2015 ; DOI: 10.1111/geb.12299). What is the accuracy of the dendrogram computed here? As overlap analyzes are based on a multidimensional space built using PCoA why not clustering species using the k-means algorithm applied to Euclidean distances in this space ?

Response 7: The dendrogram had a mean of absolute deviations (Magneville et al. 2022) of 0.096 between the trait-based distance and cophenetic distance (Added to Line 82 in results of the Supplementary material). The dendrogram quality is comparable to that of dendrograms reported in Maire et al. (2015). We consider the quality of the dendrogram acceptable for the purposes of identifying broad functional groups (high up on the dendrogram; see Supplementary Fig. S2), where the effects of any dendrogram bias on species distances is unlikely to be significant enough to mean species were put into the 'wrong' functional group. The benefit of using a dendrogram for identifying functional groups (over k-means in PCoA space, for example) is the transparency the dendrogram offers for visualising how the functional groups of the community 'fit' together. Throughout this manuscript, we have made a particular effort (using the dendrogram, wordclouds, and conditional inference tree node diagram; Figure 2, Fig. S2) to describe and visualise functional groups for the reader, to communicate our approach effectively.

Magneville, C., Loiseau, N., Albouy, C., Casajus, N., Claverie, T., Escalas, A., Leprieur, F., Maire, E., Mouillot, D. & Villéger, S. mFD: an R package to compute and illustrate the multiple facets of functional diversity. *Ecography* 1 (2022).

3) 19 groups were identified on the dendrogram but only 9 of them were kept for overlap analyzes. Even if those 9 groups gather 85% of the species, some excluded groups could have few species but a high biomass. In addition, some excluded groups (e.g. FG 5 with 24 species) have more species than selected groups (max = 20 species). It would be of interest to test for the overlap between tropical and temperate species in all groups. Related to this point, number of species per group differ between Figure 2 and Figure S2 (e.g. 133 vs 141 species for FG15). Which numbers are correct?

Response 8: We rejected the 10 other functional groups primarily because they had too few species (≥ 14 species represented in both regions being the cutoff). The exception, as pointed out, was FG 5 with 24 species, this functional group was not included in analyses as it was well represented in

Australia (18 species) but only had seven species in Japan, which were rarely encountered (shown in the bottom plot in Figure 1 below). A further reason for not modelling all functional groups was that many of the smaller groups were rarely encountered and/or large biomass species (e.g., manta ray, *Mobula birostris*). Both of these factors made fitting predictive models complicated or impossible.

There are more species per functional group in Figure S2 compared to Figure 2 in the main manuscript as Figure S2 represents the Japan-Australia transitional fish meta-community in both summer and winter and includes an additional 64 species observed in the Austral summer in Australia (see L 379-382 in main manuscript). These 64 Australian summer species were not recorded in the Australian winter biomass data used for modelling tropicalisation trends which are shown in Figure 2. We could have omitted the 64 Australian summer species from functional group creation altogether; however, as we were trying to create a fish meta-community spanning both regions and representing both seasons, we felt it appropriate to include them even without the corresponding biomass data. From a functional perspective, we were interested in generalist (species rich) and specialist (species poor) functional groups, and knowing which functional groups summer migrants joined helped build a more comprehensive picture. We have added text to Fig. S2 caption to better explain the differences outlined above.

Figure 1. Biomass trends of tropical (red) and temperate species (blue) in Australia and Japan, from different functional groups. Loess smoothers visualize trends over latitude for each thermal affinity and total biomass (dotted). Grey shade represents latitudinal spans of six zones (A-G) in each region. Transitional community zones are coded: A=Tropical Coral Reef, B=Cold-tropical Reef, C=Warm-

transitional Reef, D=Transitional Reef, E=Subtropical Reef, F=Temperate Kelp Reef, G=Cold-transitional Reef. Note the inclusion of FG 5 at the bottom which is not included in the main manuscript.

4) Overlap between tropical and temperate species is computed in a 2D space built on trait-based distance. However, such low-dimensionality spaces are often underestimating distance between species (Maire et al 2015). Please provide quality metrics of the PCoA and consider accounting for more PC axes to compute overlap.

Response 9: The first two axes explain 35% of the variance combined (added to line 429-430). Including more axes would add further explanatory information, however variance explained by additional axes drops off quickly. Furthermore, presenting overlap metrics in any more than the two dimensions as we have currently (e.g., Figure 4) becomes more confusing to interpret visually (especially given the number of functional group-thermal preference-region-zone combinations), for little additional gain in explanatory value. For reference, the percent variance explained by the first 40 PCoA axes are:

18.5816374 15.6472272 9.8651318 7.2909154 6.7891427 5.5508511
5.2233532 2.9074198 2.4252868 2.1425832 2.0621990 1.9027861
1.6977402 1.6665582 1.3977811 1.3036176 1.1684155 1.0459116
0.8931262 0.8556806 0.8111867 0.6893966 0.6301696 0.6118410
0.5657822 0.5278172 0.5007357 0.4920171 0.4751771 0.4627483
0.4506719 0.4395097 0.4323281 0.4299661 0.4218353 0.4173002
0.4116002 0.4089753 0.4035771 0.4001536

5) Overlap is computed based on a metric accounting only for presence of species. Therefore, it does not discriminate cases where both temperate and tropical species have high biomass, from those where one of the 2 species is rare. It would thus be complementary to compute a biomass-weighted similarity metric (e.g. Mean Nearest Neighbor Distance) to test whether increasing overlap based on occurrence is paired with increasing biomass of the most closely related species.

Response 10: We thank the reviewer for raising this issue and have conducted the suggested analyses, including plotting the results using the same style figure as Figure 4 in the main manuscript to aid visualisation and interpretation. We found that there was no consistent relationship between functional overlap of tropical and temperate thermal guilds (overlap of guild 99% Kernel UDs of species occurrence points in trait space) and biomass-weighted distance between thermal guilds (distance between guild centroids calculated on biomass weighted-species occurrence points in trait space). We found that the majority of functional groups (six) showed no significant relationship between functional niche overlap between thermal guilds based on species occurrences and distance between the biomass centre-points of thermal guilds. Five functional groups did not have enough datapoints to test correlation as they contained too many sites where one functional guild was missing (forcing functional overlap to zero and not allowing centroid distance to be calculated). The remaining four functional groups with significant correlations all showed that as functional overlap between thermal guilds increased, the distance between their centres of biomass decreased. This finding suggests that for some functional groups, our measure of competition between thermal guilds, using overlap of functional niches based on species occurrences, also captures competition between the most important (biomass-heavy) areas of those functional niches. However, in the context of the greater number of non-significant functional group relationships, no clear conclusion could be drawn. Given the reduced sample size when calculating distance between biomass weighted centroids (both thermal guilds required to be present at a site), and the inconsistent results, this additional analysis has been included in the supplement rather than the main manuscript text. These changes are located in the method text (supplementary L 51-64), results text (supplementary L 111-124), Table S2, Figure S6.

Minor comments:

- Line 51: this first sentence would be clearer if the driver of species redistribution was named here not in following sentence. Added “climate change driven”.
- Line 63: this sentence about functional groups could be moved after the trait-based approach presented in lines 64-67. We kept the sentence in Line 63 as it was, as this is our topic sentence and as such better for our purpose.
- Line 69: it is unclear why functional niche is defined for species with similar traits, since a trait-space could only be built if all species were described with the same traits. Do you mean functional groups gathering species with same trait values? Changed this to “trait values”, thanks for spotting this issue!
- Lines 87-90: add names of first author for ref 28, 29 and 30. Done at all appropriate instances (lines 86-92).
- Lines 143-144: specialist vs generalist generally refer to species ecology (or diet) based on the range of suitable environments (or preys eaten). Here definition is rather similar to the redundancy concept since group with many species are likely to have more species with similar trait values (although it needs to be confirmed using distance-based metrics). We followed the cited paper, which supports the context of functioning in terms of diet (Reference 37, Gajdzik, L., Aguilar-Medrano, R. & Frédérick, B. Diversification and functional evolution of reef fish feeding guilds. Ecology Letters 22, 572–582 (2019)).
- Line 149: “generally shows” is vague. How many groups have such a pattern and what is the average slope? Updated the text with specific statements about how many functional groups show which trends: “We find that the biomass of tropical species declines with increasing latitude within all but one of the nine functional groups in both regions (Fig. 2). The biomass of temperate species within each functional group shows the opposite trend in five functional groups. Functional pressure (total biomass) is maintained across latitude in two functional groups with comparable tropical and temperate biomass, but several functional groups are dominated by tropical biomass and their functional pressure declines with increasing latitude (e.g., Upper-benthic Herbivores FG6)”, lines 147-153).
- Line 187-196: this section about implications of findings could be moved to discussion, and adding names of fish species with these features would be more informative. Our paper largely integrates short methods, results, and discussion as per Nature group journal format, so we have opted to keep this text together with the findings it discusses. The fish species contained within the functional groups and thus with the various features discussed can now be seen in Supplementary Table S3.
- Line 376: given the large spatial extent of the study these are not meta-communities, rather meta-ecosystems. We are analysing fishes, and we don’t believe that a single taxon constitutes an ecosystem. To address the point that we are combining them across large spatial scales, we have placed meta-communities in inverted commas, Line 378.

REVIEWER COMMENTS

Reviewer #2 (Remarks to the Author):

The authors have revised several parts of the MS (and of Supp Materials) to address the points raised during the first review round. Especially, methods are now presented in more details.

I still have 2 methodological concerns:

The dendrogram used for the clustering of species into functional groups does have a low mean absolute deviation, meaning that on average it faithfully represents trait-based distance between species. However, the main pitfall with dendrograms is that they often inflate distance between (a few) species that actually have close trait values (e.g. top-right panel of figure 3 from Maire et al 2015). Such a bias has consequences for species clustering with functionally similar species being in distinct groups. Did you check on the Gower distance vs cophenetic distance scatterplot that there was no such bias with your set of species (at least those in the 9 key functional groups)? From what is displayed with word cloud, it seems this is not the case, but it will be better to include such a plot in the Supp Figure.

Similarly, authors justify the use of only 2 axes from the PCoA to measure overlap between thermal guilds as "variance explained by additional axes drops off quickly" while 3rd and 4th axes actually explained together as much as PC2. I agree that it is enough to display patterns along the 2 first axes but computation of indices could be done with 4 axes, and my guess is that it will even reinforce the findings.

I also suggest a few minor changes to clarify some sentences, table and figure:

Line 250: "diet is also a non-significant driver" to "Diet had no significant effect on leading-edge latitudes, with for instance..."

Line 310-312: "functional space" is confusing here, "filling of the functional space" would be more accurate

Figure 4: add a title to Y axis and add "FG" before id of functional group above fish picture.

Table S1: It would be relevant to add a column with trait type (continuous, ordinal, categorical) and to replace « numeric » in trait value for Size and depth by range of values across all species.

Responses to reviewer comments

Temperate functional niche availability not resident-invader competition shapes tropicalisation in reef fishes, Miller et al.

Reviewer #2 (Remarks to the Author):

The authors have revised several parts of the MS (and of Supp Materials) to address the points raised during the first review round. Especially, methods are now presented in more details.

I still have 2 methodological concerns: The dendrogram used for the clustering of species into functional groups does have a low mean absolute deviation, meaning that on average it faithfully represents trait-based distance between species. However, the main pitfall with dendrograms is that they often inflate distance between (a few) species that actually have close trait values (e.g. top-right panel of figure 3 from Maire et al 2015). Such a bias has consequences for species clustering with functionally similar species being in distinct groups. Did you check on the Gower distance vs cophenetic distance scatterplot that there was no such bias with your set of species (at least those in the 9 key functional groups)? From what is displayed with word cloud, it seems this is not the case, but it will be better to include such a plot in the Supp Figure.

Response 1: The Shepard diagram above shows the cophenetic distances, as represented in the dendrogram, provide a good representation of the original Gower distances, tracking the 1:1 diagonal line. The point density gradient (blue, low to yellow, high), shows that at higher distances the majority of cophenetic distances still faithfully represent their Gower counterparts despite a greater spread in Gower distances represented by a single Cophenetic distance. This plot has been added to the supplementary material (Supplementary Fig. S3).

Similarly, authors justify the use of only 2 axes from the PCoA to measure overlap between thermal guilds as “variance explained by additional axes drops off quickly” while 3rd and 4th axes actually explained together as much as PC2. I agree that it is enough to display patterns along the 2 first axes but computation of indices could be done with 4 axes, and my guess is that it will even reinforce the findings.

Response 2: We thank the reviewer for suggesting this, as it has indeed reinforced the findings. To extend the analyses to include the extra two axes we created six two-dimensional representations of functional space through all pairwise combinations of the first four PCoA axes. Functional niche metrics are calculated within each functional space and then averaged together using the variance explained by each space’s PCoA axes pair as a weight. The results are minor changes to the functional overlap and filtering values and four changes to associated significance in the Kendall correlations (Table 1). Now FG 10 in *both* regions show decreasing tropical biomass and increasing functional niche overlap and FG 4 in Japan no longer shows significant correlation between increasing tropical biomass and increasing functional niche overlap – both of these changes align better with the trends visualised in Figure 4. Method text has been added to L430-435, and results updated in L210-214, L246-248, and Table 1.

I also suggest a few minor changes to clarify some sentences, table and figure:

Line 250: “diet is also a non-significant driver” to “Diet had no significant effect on leading-edge latitudes, with for instance...”

Response 3: Done, thank you.

Line 310-312: “functional space” is confusing here, “filling of the functional space” would be more accurate

Response 4: Done.

Figure 4: add a title to Y axis and add “FG” before id of functional group above fish picture.

Response 5: Done.

Table S1: It would be relevant to add a column with trait type (continuous, ordinal, categorical) and to replace « numeric » in trait value for Size and depth by range of values across all species.

Response 6: Done.

REVIEWER COMMENTS

Reviewer #2 (Remarks to the Author):

The revised version has not fully addressed the 2 methodological concerns that could affect the findings.

The plot illustrating trait-based distance versus dendrogram-based distance confirms that some species pairs with the most similar trait values (Gower distance <0.2) are among the furthest from each other on the dendrogram (cophenetic distance >0.4).

Hence, some functionally similar species could be part of different groups.

Could you check that none of the species pairs with Gower <0.2 are in different groups?

If this is the case, then I suggest to use a k-means clustering in the 4D-functional space used for other analyzes.

My previous advice to run analyzes accounting for the 4-axes of the functional space was misunderstood. I meant running computation with all 4 axes simultaneously, not averaging values from computation for the 6 pairs of 2 axes. Indeed, the 2D overlap accounting for PC3 and PC4 have the same pitfalls than the one accounting for only PC1-PC2. For instance, in a 3D case, if overlap on PC1 and PC2 is 90% but species from the 2 assemblages fill distinct ranges along PC3 then the average of the 3 2D-overlap $((90+0+0)/3=30\%)$ is misleading as actual overlap in 3D is 0% because of the non-overlap along PC3.

R packages kde and TPD can deal with 4 dimensions to compute overlap between kernels.

Response to Reviewer comments round 3, 18 Jan 2023

Temperate functional niche availability not resident-invader competition shapes tropicalisation in reef fishes, Miller et al.

REVIEWER COMMENTS

Reviewer #2 (Remarks to the Author):

The revised version has not fully addressed the 2 methodological concerns that could affect the findings.

The plot illustrating trait-based distance versus dendrogram-based distance confirms that some species pairs with the most similar trait values (Gower distance <0.2) are among the furthest from each other on the dendrogram (cophenetic distance >0.4).

Hence, some functionally similar species could be part of different groups.

Could you check that none of the species pairs with Gower <0.2 are in different groups?

If this is the case, then I suggest to use a k-means clustering in the 4D-functional space used for other analyzes.

Reply 1: There is only one species, *Cantheschenia grandisquamis*, that displays Gower distance <0.2 with some other species, but occurs in different functional groups from these species. Please see Figs 1-3 below which demonstrate where these distance pairs occur. *Cantheschenia grandisquamis* was placed in FG8 (Benthic Herbivore/Omnivores), but there are 140 species for which it has Gower distances <0.2 which were placed in FGs 15 (Benthic Predators), 16 (Corallivores) and 4 (Benthic Planktivores). It was hard for our dendrogram analysis to place *Cantheschenia grandisquamis*, because it is the only species with an NA in the Diet trait. Gower distances for *Cantheschenia grandisquamis* were therefore mostly driven by a “benthic” Habitat Association trait which was shared between *Cantheschenia grandisquamis* and the 140 similar species. As there was only one species potentially placed into the incorrect functional group by the dendrogram analysis, and this species was data limited, we maintain that the dendrogram provides an acceptable representation of the original Gower distances and is fit for purpose.

Fig1. The Shepard diagram included in the supplementary material in the last round of revision.

Fig 2. Version of Fig. 1 filtered to show species pairs with Gower distance <0.2. Black points show species pairs where both species were assigned the same functional group. Coloured points show where the species pair occurred in different functional groups.

Fig 3. Version of Fig. 1 filtered to show species pairs with Gower distance <0.2. Points are coloured by the 140 species pairs where species were assigned to different functional groups, note all species pairs feature *Cantheschenia grandisquamis*.

My previous advice to run analyzes accounting for the 4-axes of the functional space was misunderstood. I meant running computation with all 4 axes simultaneously, not averaging values from computation for the 6 pairs of 2 axes. Indeed, the 2D overlap accounting for PC3 and PC4 have the same pitfalls than the one accounting for only PC1-PC2. For instance, in a 3D case, if overlap on PC1 and PC2 is 90% but species from the 2 assemblages fill distinct ranges along PC3 then the average of the 3 2D-overlap $((90+0+0)/3=30\%)$ is misleading as actual overlap in 3D is 0% because of the non-overlap along PC3.

R packages `kde` and `TPD` can deal with 4 dimensions to compute overlap between kernels.

Reply 2: We thank the reviewer for clarifying, and have tried both R packages to calculate functional overlap in 4-axes simultaneously. Unfortunately, we have not been able to successfully run the analyses across a sufficient number of Sites to make a meaningful interpretation. This is due to a sample size issue: all functional groups have many more tropical species than temperate species (Figure 2). We were typically able to construct 4-dimensional Trait Probability Density functions for relatively species-rich tropical thermal guilds within most functional groups, but could rarely create the counterpart TPD function for temperate thermal guilds, with which to compute overlap.

As we compute functional overlap for communities at Zone and Site level (main manuscript Figure 4; Figure S1), rather than the entire transitional gradient community, the actual number of species per thermal guild represented in functional niche overlap analyses is less than the functional group totals shown in main manuscript Figure 2. This meant for the majority of Zone and Site level analyses, we did not have the sample size to make 4-dimensional TPDs for the temperate thermal guild and could therefore not calculate functional overlap.

Our study requires calculation of functional overlap at Site level to have enough sampling power to test for correlation with tropicalisation trends. With our inability to compute simultaneous 4-dimensional functional overlap we followed the reviewer's example, and revisited the original averaged 6-pair 2-dimensional approach to identify any pairs of axes with zero overlap that when averaged with other axes would incorrectly return a non-zero value (i.e., implying some overlap). We found that there were instances of this occurring, although uncommon (out of 25 sites in Australia, FG6 at 4 sites, FG1 at 3 sites, FG4 at 2 sites, and FG15 at 1 site; and out of 28 sites in Japan, FG4 at 5 sites, FG1 at 3 sites, and FG8 at 1 site). We have now forced these instances to return a zero, non-overlap result and recalculated the results (updated in L210-214, L246-248, and Table 1) and updated methods (L446-448). Given the few FGs and sites affected this has had minimal impact on results (minor changes to decimal places for the aforementioned FGs in Table 1).

We would like to note the usefulness of the Carmona *et al.* (2016) approach and the TPD R package, and had this project had sufficient sample size, the greater insight on functional overlap returned from the TPD method compared to boundary-based method used here would have been beneficial. However, we also feel that given the breadth of content in this study (tropicalisation trends across eight functional groups, over latitudinal gradients in two regions), of which functional overlap between thermal guilds is only one component, the reviewer will accept the non-simultaneous calculation of 4-dimensional functional overlap given our inability to implement the simultaneous method.

REVIEWERS' COMMENTS

Reviewer #2 (Remarks to the Author):

In the response letter, the authors have answered in details to the 2 questions about their methodological choices.

First, the authors found that the only large mismatches between trait-based Gower distance and cophenetic distance on the dendrogram are due to a single species (*Canthoeschenia grandisquamis*) with missing value for diet.

The presence of missing values in the trait dataset should be mentioned in the Methods section of the paper (as well as their low frequency) as it influences Gower metric properties (i.e. distance between a species pair is computed only on the subset of traits with no missing values) and eventually the dendrogram. The issue with *C grandisquamis* may also be mentioned in the Supp Mat.

Second, the authors clarified that is not possible to compute overlap in 4D between tropical and temperate species due to the low number of temperate species in most sites. It is now mentioned in the Methods that when overlap on an axis was null, the multidimensional overlap metric was set to 0 not to average among the 6 pairs.